# The Impact of Weir Construction in Korea’s Nakdong River on the Population Genetic Variability of the Endangered Fish Species, Rapid Small Gudgeon (*Microphysogobio rapidus*)

**DOI:** 10.3390/genes14081611

**Published:** 2023-08-11

**Authors:** Yang-Ki Hong, Kang-Rae Kim, Keun-Sik Kim, In-Chul Bang

**Affiliations:** 1Natural History Division, National Science Museum, Daejeon 34143, Republic of Korea; ykhong17@korea.kr; 2Animal & Plant Research Department, Nakdonggang National Institute of Biological Resources, Sangju 37242, Republic of Korea; kimkangrae9586@gmail.com; 3Restoration Research Team (Fishes/Amphibians & Reptiles), Research Center for Endangered Species, National Institute of Ecology, Yeongyang-gun 36531, Republic of Korea; kskim@nie.re.kr; 4Department of Biosystem, Soonchunhyang University, Asan 31538, Republic of Korea

**Keywords:** *Microphysogobio rapidus*, microsatellite, endangered fish

## Abstract

*Microphysogobio rapidus*, an endemic cyprinid fish species found exclusively in Korea, has been identified in only two tributaries of the Nakdong River. The species predominantly occupies the near-gravel bottom waters within shallow sections of the middle and lower reaches of the river, characterized by swift currents. *M. rapidus* is currently recognized as a critically endangered species due to its distinct habitat preference, as well as the negative impacts of stream dam development and water environment pollution. In this study, we used 10 microsatellite markers to examine the genetic diversity of *M. rapidus* in the upper Nam (UN), lower Nam (LN), and Deokcheon Rivers (DC) in Korea, with a specific focus on assessment of the impact of dam development. Fish sampled from the UN and LN showed a greater average number of alleles and allelic richness (*A* = 18.3–18.4, *A*_R_ = 13.8) compared to those from DC (*A* = 11.8, *A*_R_ = 11.5). The observed heterozygosity among the fish examined ranged from *H*_O_ = 0.748 (LN) to 0.766 (DC). All three fish groups exhibited a significant departure from Hardy–Weinberg equilibrium (HWE) (*p* < 0.05). Despite having the largest effective population size (*N*_e_ = 175 and 157, respectively), the fish sampled from UN and LN showed the highest inbreeding coefficients (*F*_IS_ = 0.056–0.053, respectively), which were highly significant (*p* < 0.01). In contrast, the fish sampled from DC exhibited the smallest effective population size (*N*_e_ = 61) and showed an inbreeding coefficient close to zero (*p* > 0.05). BOTTLENECK analysis and estimated M-ratio values (0.341–0.372) revealed indications of past population size reduction in all fish groups examined. No significant genetic differentiation (*F*_ST_ < 0.05) was detected using the DAPC, STRUCTURE, and AMOVA among the fish studied. However, pairwise comparisons of *F*_ST_ between fish sampled from the Nam and Deokcheon Rivers revealed significant values (*p* < 0.001) ranging from 0.013 to 0.014. In addition, the closest genetic distance (0.026) was observed between UN and LN, while the greatest distance (0.087) was found between UN and DC. Analysis of gene flow rates among the fish examined indicated asymmetrical gene exchange within the Nam River, which was 31.51% in the downstream direction (from UN to LN), with a minimal gene flow rate (0.41%) in the upstream (from LN to UN) direction. The opposite trend was recorded between DC and LN, with a higher gene flow rate (29.74%) in the upstream direction compared to the downstream direction (0.12%). Our study highlighted the importance of implementing long-term conservation efforts focused on maintaining river integrity by removing water barriers such as weirs that impede fish migration and implementing active protection measures, such as aquaculture breeding and reasonable stocking practices, to preserve *M. rapidus* in the study area.

## 1. Introduction

A total of 34,012 weirs have been constructed in rivers in Korea to ensure an adequate supply of agricultural water, specifically for rice cultivation. However, structures such as dams, weirs, and other manmade obstacles in streams significantly impact the migration patterns of fish species due to habitat fragmentation, resulting in reduced fish population size, restricted gene flow, and a consequent decrease in genetic diversity, ultimately leading to increased levels of homozygosity [1,2,3,4,5]. The reduction in fish population size resulting from habitat fragmentation can exacerbate inbreeding and genetic drift within fish populations, leading to diminished prospects of successful reproduction and survival, thereby significantly increasing the long-term risk of extinction [6].

*Microphysogobio rapidus* is an endemic cyprinid fish species found exclusively in Korea, which inhabits fast-flowing shallow waters with gravel bottoms in the middle and lower reaches of the Nam and Deokcheon rivers, tributaries of the Nakdong River [7,8]. *M. rapidus* is currently classified as a “critically endangered” species [9] due to its small population size, limited distribution, and high vulnerability to human-induced environmental alterations. Recognizing the urgency of its conservation, the Ministry of the Environment designated this species for protection in 2012, classifying it as Class I endangered wildlife [10]. At present, the habitat of *M. rapidus* faces significant fragmentation, with 35 weirs installed in the Nam River and an additional 39 weirs in the Deokcheon River, contributing to the substantial division of its habitat. Furthermore, the presence of Jinyang Lake acts as a further barrier separating the two rivers [11]. Given the uncertain impact of the weirs and the presence of Jinyang Lake on *M. rapidus*, it is important to conduct studies focused on genetic diversity and structure. Obtaining such data will be crucial for the development of a systematic conservation strategy, which is urgently required to safeguard the future of this species.

Despite a few molecular phylogenetic studies, the lack of efficient molecular markers and comprehensive population genetic studies has hindered the comprehensive evaluation of the genetic diversity and genetic structure of this species [8,12,13,14,15,16]. DNA fingerprinting analysis, including the utilization of microsatellite markers, is a widely used molecular method for evaluating genetic variability in natural populations, including endangered fish species [17,18]. Microsatellites, which exist in all genomes, exhibit high levels of polymorphism and follow codominant inheritance patterns, making them ideal markers that can be easily genotyped using PCR-based techniques [19].

This study was performed to analyze the genetic diversity and structure of the *M. rapidus* population in the Nam and Deokcheon rivers using microsatellite markers. The obtained genetic data can be utilized to develop an effective conservation strategy for the existing shoal population.

## 2. Materials and Methods

### 2.1. Sample Collection

As *M. rapidus* has been designated as Class I endangered wildlife by the Ministry of Environment, proper permissions for capture were obtained from the Nam River (permit no. 2012-07, no. 2013-02) and Nakdong River Basin Environmental Offices of the Ministry of Environment in 2012 and 2013. The study area consisted of one sampling site in the Deokcheon River (DC) and two sites along the Nam River, i.e., the upper Nam (UN) and lower Nam (LN), both of which are tributaries of the Nakdong River before it reaches Jinyang Lake (Figure 1). To collect specimens for the study, a cast net with a mesh size of 16 mm (covering an area of 4.5 m^2^) and a scoop net with a mesh size of 8 mm (covering an area of 1.35 m^2^) were employed, and we collected a total of 120 specimens, i.e., NAM_UN01–48 from UN, NAM_LN01–48 from LN, and DC_Rv01–24 from DC. After capture, individuals were anesthetized with MS-222 (Syndel, Nanaimo, BC, Canada), following which a portion of the pelvic fin was sampled and preserved in absolute ethanol for further analysis in the laboratory. Subsequently, the captured fish were immersed in 50 ppm oxytetracycline solution and released back into the natural habitat.

### 2.2. Genomic DNA Extraction

After transportation to the laboratory, the fin samples were washed thoroughly with ethanol, rinsed with distilled water, and treated with TNES-urea buffer containing 8 M urea, 10 mM Tris-HCl (pH 7.5), 125 mM NaCl, 10 mM EDTA, and 1% SDS for DNA isolation, according to the methodology described previously by Asahida et al. [20]. The quantity and quality of extracted genomic DNA (gDNA) were determined spectrophotometrically (Nanodrop-ND1000; Thermo Fisher Scientific, Waltham, MA, USA).

### 2.3. Microsatellite Marker Development

#### 2.3.1. DNA Cutting and Ligation

Microsatellite DNA was isolated as described previously [21]. The extracted gDNA (1 μg) was digested with *Rsa*I (New England Biolabs, Ipswich, MA, USA) for 10 s according to the manufacturer’s instructions. The resulting DNA fragments were treated with mung bean nuclease (New England Biolabs) for 30 min to obtain blunt ends and dephosphorylated using calf intestinal phosphatase (New England Biolabs). DNA fragments of 200–800 bp were separated using electrophoresis on 1.5% agarose gels and recovered using a QIAquick gel extraction kit (Qiagen, Germantown, MD, USA). The recovered DNA fragments were ligated to adapters (SNX/SNX reverse linker) by combining with 60 μM SNX adapter, 5 μL of NEB #2 buffer, 0.5 μL of 100× BSA, 1 μL each of *Nhe*I (New England Biolabs) and *Xmn*I (New England Biolabs), 50 mM rATP (Promega, Madison, WI, USA), and 2000 units of ligase (New England Biolabs) in a total volume of 50 μL.

#### 2.3.2. Enrichment of Microsatellite DNA Libraries

The microsatellite DNA regions of interest were selectively recovered using a Magenesphere magnetic separation kit (Promega) and biotin-labeled probes (GT)_10_ and (CT)_10_. After the addition of 50 μL of hybridization solution (12× SSC, 0.1% SDS) together with the probes to the linker-ligated DNA, the mixture (100 μL) was heated to 95 °C for 15 min and then incubated at 60 °C for 12 h. Hybridization was performed using Streptavidin paramagnetic particles (Promega) in accordance with the manufacturer’s instructions. Single-stranded DNA was eluted at 95 °C for 15 min in 100 μL of low-TE buffer containing 10 mM Tris-HCl (pH 8.0) and 0.1 mM EDTA. PCR was performed in mixtures consisting of 10 μL of DNA, 4 μL of 10 μM SNX primer, 0.3 μL of Vent (-exo) polymerase (New England Biolabs), 4 μL of dNTP mix, 5 μL of 10× Thermopol buffer, and 26.7 μL of sterilized distilled water with denaturation at 96 °C for 5 min followed by 40 cycles of 96 °C for 45 s, 62 °C for 1 min, 72 °C for 2 min, and a final elongation step at 72 °C for 5 min.

#### 2.3.3. Cloning and Selection

The double-stranded DNA fragments were cut with *Nhe*I, and the pUC18 vector was cut with *Xba*I. After dephosphorylation with calf intestinal phosphatase, aliquots of 100 ng of each DNA fragment and vector were mixed with 2 μL of NEB #2 buffer, 0.2 μL of 100× BSA, 1 μL of *Nhe*I, 2 μL of 10 mM rATP (Promega), and 400 units of ligase (New England Biolabs) in a total volume of 20 μL. The reaction mixtures were incubated at 16 °C for 30 min and 37 °C for 15 min for a total of 40 cycles. The ligated vector was transformed into XL1-blue MRF strain (Stratagene, La Jolla, CA, USA) using cold shock, spread on LB agarose plates containing 100 μL of 10 mM IPTG and 100 μL of 2% X-Gal, and incubated overnight at 37 °C. White colonies, expected to contain microsatellite regions, were subjected to PCR using the two probes and the M13 forward/reverse primers. The amplified products were confirmed using 1.5% agarose gel electrophoresis. The respective positive colonies were cultured in LB medium, and the microsatellite-containing vector was isolated using a QIAprep Spin Miniprep kit (Qiagen) and sequenced by Macrogen (Seoul, Republic of Korea). Finally, primer sequences suitable for PCR were generated from the upstream and downstream sequences of the microsatellites using PRIMER 3 software ver. 0.4.0 [22].

### 2.4. PCR and Genotyping

Six previously reported microsatellite markers [23] and four markers newly developed in this study were amplified using the PCR protocol established by Kim [23]. Information on the markers is provided in Table 1. For PCR, 20-μL reaction mixtures containing 20 ng of genomic DNA and 5 μM of the forward fluorescent (6-FAM, HEX, NED) and reverse primers for each marker were prepared using an Accupower^®^ PCR premix kit (Bioneer Inc., Daejeon, Republic of Korea) (Table 2). The PCR conditions were as follows: denaturation at 94 °C for 5 min, followed by 35 cycles of 94 °C for 1 min, 60 °C for 30 s, 72 °C for 1 min, and a final elongation step at 72 °C for 7 min. The amplified product was confirmed by 1.5% agarose gel electrophoresis, and each marker band was diluted to a concentration appropriate for genotyping. The diluted PCR products were mixed with Genescan™ 400HD (ROX) size standard and the HiDi mixture and then denatured at 95 °C for 5 min. Genotyping was performed on an ABI 3730xl DNA analyzer (Applied Biosystems, Foster City, CA, USA) as specified by the manufacturer.

### 2.5. Genetic Diversity and Structure Analysis

The precise size of all alleles was determined using Peak Scanner™ software ver. 1.0 (Applied Biosystems). The genotypic data were validated using MICRO-CHECKER software ver. 2.2.3 [24], which facilitated the identification of potential instances of null alleles, large allele dropouts, scoring errors, and input errors that may have occurred during data acquisition.

The number of alleles (*A*) and observed heterozygosity (*H*_O_) and expected heterozygosity (*H*_E_) were calculated using Cervus software ver. 3.0 [25]. Samples with differences between populations in the number of individuals sampled can be analyzed using FSTAT software ver. 1.2 to estimate the allelic richness in cases where PCR amplification cannot be performed due to insufficient amounts of gDNA from an endangered species [26]. Genepop software ver. 4.0 [27] was utilized for the analysis of deviation from Hardy–Weinberg equilibrium (HWE), and the results were compared with those generated using the Markov chain method. The inbreeding coefficients (*F*_IS_) within the fish sampled from LN, UN, and DC were calculated using Arlequin software ver. 3.5 [28], with significance determined through 10,000 repetitions. The effective population size for each sampled fish group was determined using the linkage disequilibrium method using LDNe software ver. 1.0 [29].

Analysis of genetic differentiation (*F*_ST_) among the examined fish groups was conducted using Arlequin software ver. 3.5 [28] with 10,000 repetitions. Genetic distance between populations was measured using the method described previously [30] and TFPGA software ver. 1.3 [31]. Migration rates between the examined fish groups were analyzed using BayesAss software ver. 3.0 [32]. Genetic structure clustering was assessed using a Bayesian model in STRUCTURE software ver. 2.3 [33]. To estimate the most likely number of genetic clusters (*K*), from 1 to 5 clusters were simulated under an admixture model. Moreover, a non–model-based genetic clustering method, Discriminant Analysis of Principal Components (DAPC), was conducted using the R package ADEGENET ver. 2.1.3 [34], to examine the genetic structure and assess genotype distribution among the fish groups included in the analysis.

## 3. Results

### 3.1. Development of Microsatellite Markers

The microsatellite markers used for population genetic analysis of *M. rapidus*, including those newly developed in the present study, are listed in Table 1. The microsatellite markers were developed by selecting 400 colonies, of which 29 were sequenced. Twelve sequences were available for the construction of primers that could specifically amplify the microsatellite region. Following PCR amplification with the 12 primer pairs created using the base sequences, bands of the expected sizes from eight markers were effectively amplified at an annealing temperature of 58 °C. However, due to the high probability of null alleles, only four markers were retained for further analysis: MRms245-1, MRms245-2, MRms245-3, and MRms637.

### 3.2. Genetic Diversity and Population Structure

The results of the analysis based on the 10 microsatellite markers are shown in Table 2. A total of 227 alleles were detected in all fish examined, and the average number of alleles ranged from 4 (PNms172) to 44 (GBms157). Fish sampled from UN and LN showed a higher average number of alleles and allelic richness (*A* = 18.3–18.4, *A*_R_ = 13.8) compared to those from DC (*A* = 11.8, *A*_R_ = 11.5). The observed (*H*_O_) and expected heterozygosity (*H*_E_) among the fish examined ranged from *H*_O_ = 0.748 (LN) to 0.766 (DC) and from *H*_E_ = 0.775 (DC) to 0.809 (UN), respectively.

Analysis using Fisher’s exact test indicated that all three fish groups exhibited significant departure from HWE (*p* < 0.05), mainly due to heterozygosity deficiency. The microsatellite markers with the greatest and significant (*p* < 0.05) HWE departure were MRms245-1 and GBms157 in UN, MRms245-1 in LN, and GBms157 in DC.

Despite having the largest effective population size (*N*_e_ = 175 and 157, respectively), the fish sampled from UN and LN showed the highest values of the inbreeding coefficient (*F*_IS_ = 0.056 and 0.053, respectively; *p* < 0.01). In contrast, the fish sampled from DC showed the smallest effective population size (*N*_e_ = 61) and inbreeding coefficient values close to zero (*p* > 0.05). BOTTLENECK analysis and estimated M-ratio values (0.341–0.372) revealed signs of past population size reduction in all fish groups examined (Table 3).

Pairwise comparisons of genetic differentiation (*F*_ST_) between fish sampled from the Nam and DC Rivers revealed very low but significant values (*p* < 0.001), ranging from 0.013 to 0.014. The closest genetic distances (0.026) were observed between UN and LN, while the greatest distance (0.087) was found between UN and DC (Table 4). Analysis of molecular variance (AMOVA) between sampling sites in the Nam River (UN, LN) and Deokcheon River (DC) revealed a variance of 1.20% between groups and 98.71% within a group (Table 5). The results of AMOVA within the water system of the Nakdong River (LN, UN, DC) indicated a between-group variance of 0.69% and a within-group variance of 99.31%. These findings suggested very low levels of genetic differentiation between UN, LN, and DC, indicating that they belong to the same population.

DAPC, STRUCTURE, and AMOVA did not detect any significant genetic differentiation within the fish studied (Figure 2). The differences were too small to determine the appropriate *K* (Figure 2). In the DAPC results obtained using the non–model-based method, UN and LN were closely associated, while DC appeared to be partially divided (Figure 3). However, as shown in Figure 4, the distribution of genotypes indicated a single population, suggesting the presence of mutually shared genotypes.

Analysis of gene flow rates among the fish examined using BayesAss software indicated asymmetrical gene exchange within the Nam River that was 31.51% in the downstream direction (from UN to LN), with a minimal gene flow rate (0.41%) in the upstream (from LN to UN) direction. Opposite trends were recorded between DC and LN, with a higher gene flow rate (29.74%) recorded in the upstream direction compared to the downstream direction (0.12%). A near lack of gene flow (0.06–0.16%) was recorded between UN and DC (Figure 5).

## 4. Discussion

In this study, 10 microsatellite markers were used for population genetic analysis of the critically endangered fish species, *M. rapidus*. The average number of alleles recorded in the present study was *A* = 22.7, which was comparable to or greater than the average number of alleles detected in *Pseudopungtungia nigra* (*A* = 14.4), *Pseudopungtungia tenuicorpa* (*A* = 19.2), *Gobiobotia macrocephala* (*A* = 22.4), *Gobiobotia brevibarba* (*A* = 18.8), and *Gobiobotia naktongensis* (*A* = 21.5) [23]. The average expected heterozygosity for genetic diversity at the species level was 0.804, which exceeded the values found in other endangered fish of the same subfamily. It was also higher than the values observed in *G. brevibarba* and *G. macrocephala*, but lower than those recorded in *P. nigra*, *P. tenuicorpa*, and *G. naktongensis* [23]. These observations suggested that the endangered status of *M. rapidus* has not significantly impacted its genetic diversity. However, the species remains at risk of extinction due to human activities other than weir construction [35]. The effective population size was determined to be 177 (122–296), which is higher than the effective population size of 100 required to maintain the size of the population in the short term but far less than the effective population size of 1000 required over the long term [36]. In conservation genetics, a small population size potentially accelerates extinction processes by extinction factors [37]. One important genetic factor that can affect extinction risk at small population sizes is inbreeding depression [36]. Based on this rationale, it can be assumed that the long-term sustainability of *M. rapidus* is at risk [36,37]. In addition, the low but significant (*p* < 0.001) inbreeding index of the entire population (0.044) indicated that genetic drift in the population could lead to rapid loss of genetic diversity.

The fish in the DC group exhibited lower genetic diversity, reduced average number of alleles, and lower allelic richness compared to those in the LN and UN groups. Moreover, the average effective population size for the DC subpopulation was 61, which falls below the threshold size of 100 required for the maintenance of the population size in the short term [6].

In the analysis of genetic differentiation among sampled fish, the DC group exhibited small but significant genetic differentiation when compared to the UN and DC groups. However, the results of STRUCTURE, DAPC, and AMOVA strongly supported that DC, LN, and UN represent a single genetic cluster. The existence of a small number of subpopulations resulting from low genetic differentiation is considered a significant risk to the continued survival of a species [38]. Having limited subpopulations with low genetic diversity can make the species more vulnerable to various threats, increasing the likelihood of its decline or extinction over time. In the case of *M. rapidus*, there are three sites but only one population. Therefore, although genetic diversity is high, it is likely to become extinct due to habitat destruction or climate change, considering that the population has decreased to the point where it cannot actively respond to environmental changes evolutionarily.

BayesAss-based genetic flow analysis, conducted to estimate genetic flow over the short term (5–10 generations), revealed a lack of gene exchange between the DC group and the UN and LN groups. In the case of fish such as *Thymallus thymallus*, *Salvelinus leucomaenis*, and *Lethenteron* sp., it has been reported that manmade structures such as weirs, sluice gates, and dams installed in rivers impede migration [3,4,5]. A fishway installed on the right side of the berm cannot be used by the small *M*. *rapidus* due to its high slope. Fish frequently migrate during the spawning season, but the spawning season of *M*. *rapidus* is in the dry season (4–5 months), such that the low water level will hinder the migration of fish in the Nam River (UN, LN) to those in the Deokcheon River (DC) via Duin weir. Genetic flow from upstream to downstream is common. Genetic flow from the DC to LN population was 29.74%, indicating an upstream direction in short generations. This confirmed unidirectional genetic flow from DC to LN based on the detection of some genotypes in the gene structure. LN and UN showed little gene flow over short generations, despite the absence of artificial structures blocking gene flow. However, given the genetic diversity, there may have been gene flow in the past rather than recently, but this was not confirmed in this study. Therefore, further analyses of genetic flow are necessary.

## 5. Conservation Implications

The fact that the DC, LN, and UN groups constitute a single population implies that there is only one remaining population of *M. rapidus*, leading to a high risk of extinction and confirming its critically endangered status.

The upstream genetic flow between DC and LN has a flow rate of 29.74%, but the upstream flow between LN and DC appears to be almost nonexistent (0.12%). The Duin weir is located between LN and DC. Considering the results of genetic flow analyses, Duin weir (2.0 × 343 × 1.5 m, 1945 year) appears to be blocking the movement of fish upstream of the shoal. A terraced fishing ground (4.5 × 36 × 1.5 m) was completed on the right side of Duin weir in 2011, but it is estimated that its use is limited. There are many weirs in DC. In particular, the Duin weir is judged to have intensified the blockage of gene flow. To ensure the effective conservation of the species, measures should be taken to facilitate genetic exchange across the study area.

Our study highlighted the crucial importance of implementing long-term conservation efforts focused on maintaining river integrity by removing water barriers such as weirs that impede fish migration and implementing active protection measures, such as aquaculture breeding and reasonable stocking practices, to preserve *M. rapidus* in the study area.

## Figures and Tables

**Figure 1 genes-14-01611-f001:**
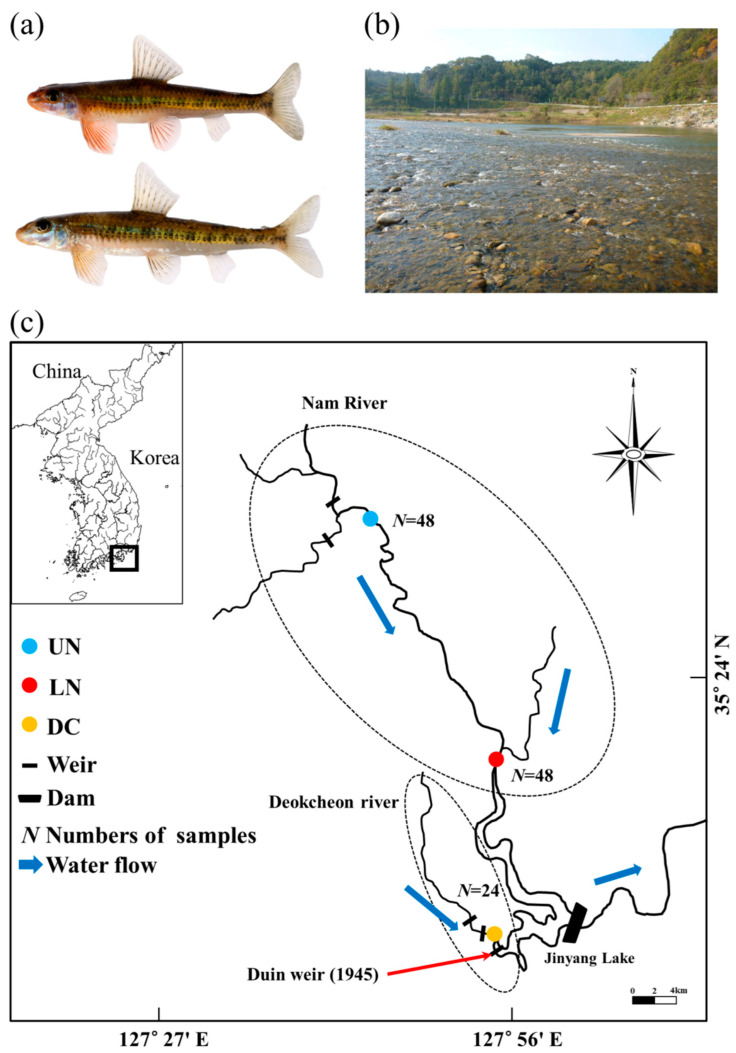
(**a**): Male (**top**) and female (**bottom**) *Microphysogobio rapidus*. (**b**): *M*. *rapidus* and its habitat. (**c**): *M*. *rapidus* from the Nam and Deokcheon river tributaries in Republic of Korea.

**Figure 2 genes-14-01611-f002:**
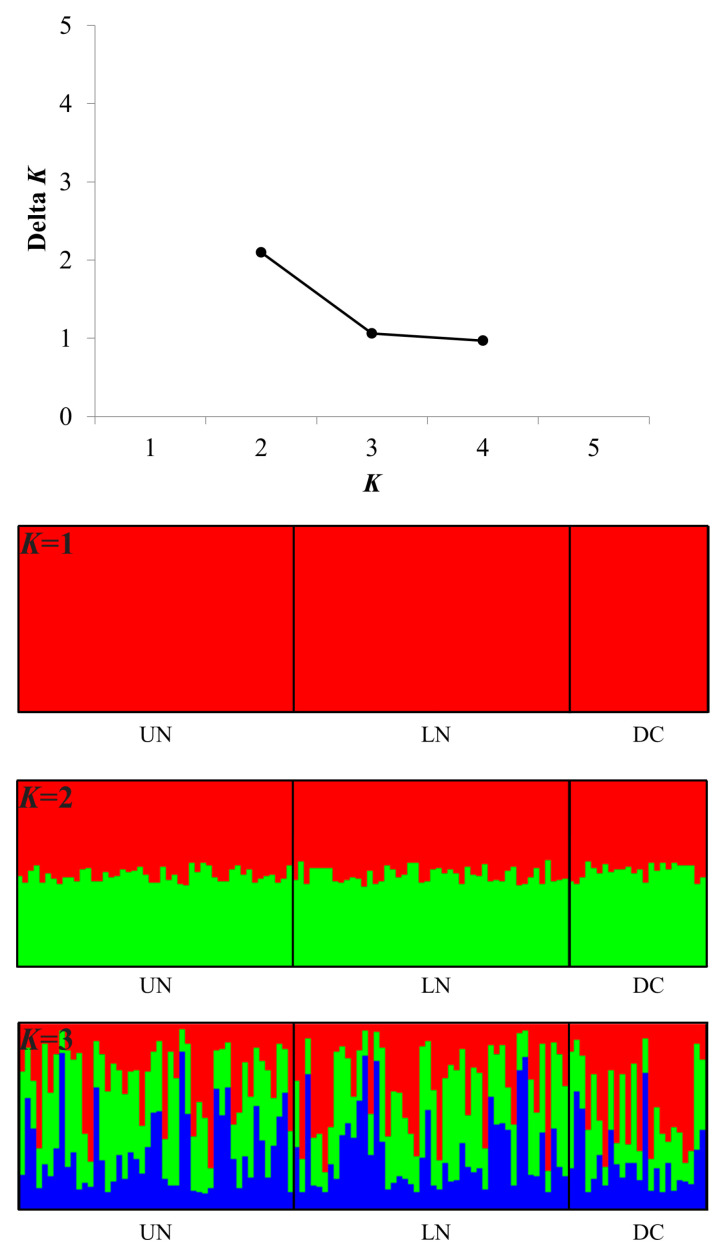
Genetic structure plot of the UN, LN, and DC groups of *M. rapidus* for putative *K* = 1–3. Each vertical bar denotes one individual.

**Figure 3 genes-14-01611-f003:**
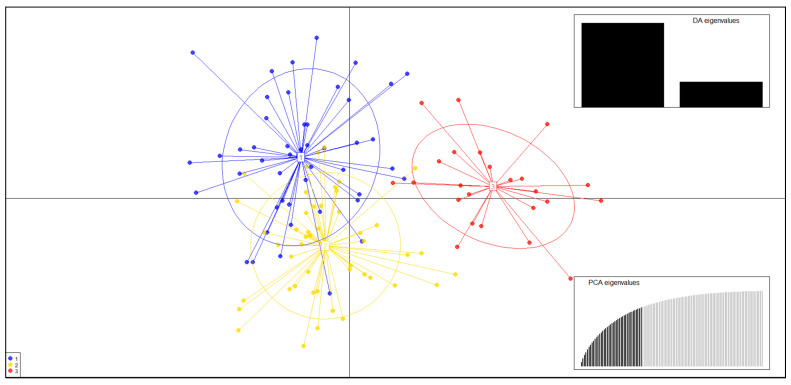
Genetic population plot based on the results of a discriminant analysis of principal components (DAPC) of *M*. *rapidus*. 1, Saengcho population (UN); 2, Danseong population (LN); 3, Deokcheon population (DC).

**Figure 4 genes-14-01611-f004:**
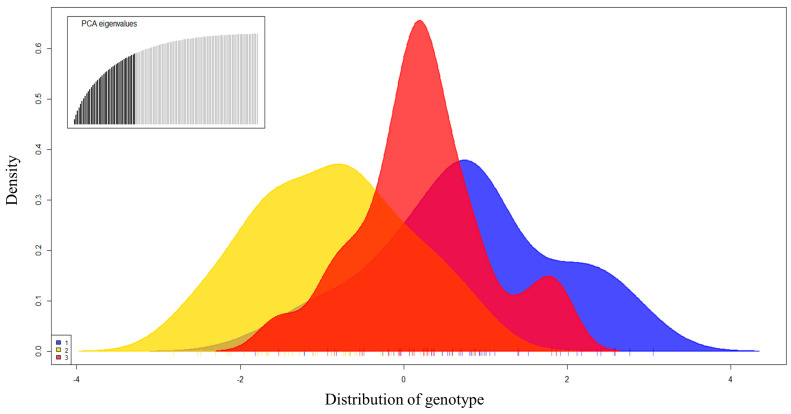
Genotype distribution according to the DAPC results for *M*. *rapidus* populations. 1, Saengcho population (UN); 2, Danseong population (LN); 3, Deokcheon population (DC).

**Figure 5 genes-14-01611-f005:**
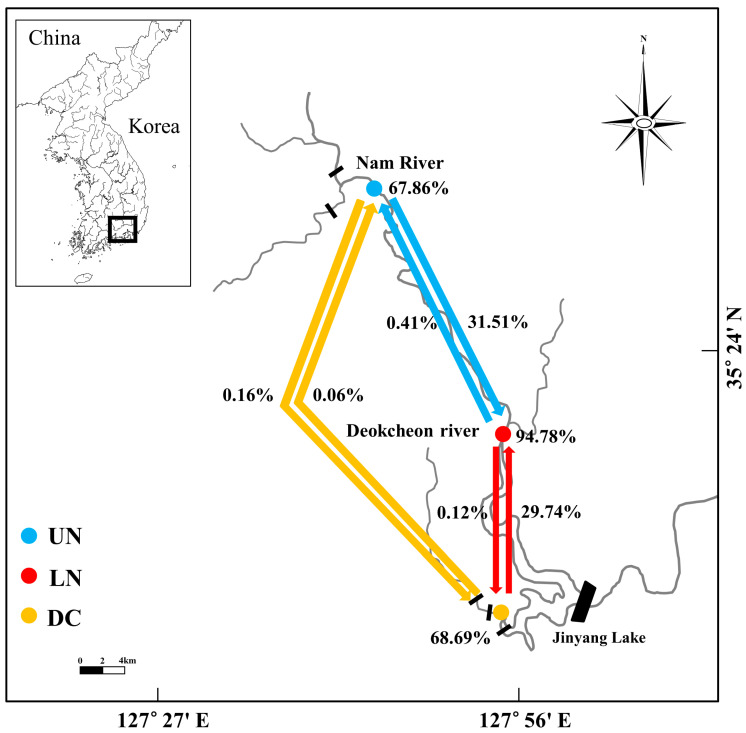
Migration rates between the UN, LN, and DC groups of *M*. *rapidus* in the Nam River basin at Gyeongsangnam-do, Korea. Arrows, smooth migration rates; black bars vertically crossing the river, weirs; black parallelograms, dams; circles, sampling stations; DC, Deokcheon River (Sugok); LN, lower Nam River (Danseong); UN, upper Nam River (Saengcho).

**Table 1 genes-14-01611-t001:** Ten microsatellite markers and primers used to study *M*. *rapidus*.

Locus	Motif Repeats	Primers (5′ → 3′)	*Ta* (°C)	Label	References
PNms172	(GT)_11_	F: TCAACCTTGTGAGTGTATGTGAGGA R: GGCGAGACACAGGCCTCTTA	56	HEX	[23]
MKms205	(CA)_12_	F: TCAGCAGGTCCACAGCTTGC R: CAGTGAGCTCCCATTTACTGTGAC	58	NED	
MKms633	(CA)_16_	F: TCGTAATTTACCCCAGCACAACA R: TTCATTTCGCCACCGAAAAA	58	6-FAM	
MKms747	(CA)_19_	F: CGAAATAATCCGCTCCCTGG R: CCCACAGGACTTTCCCTCTTG	58	HEX	
GBms157	(CA)_6_ (CA)_16_	F: GGAGTTATGAACGATAGAGCAGAC R: GAGCCTCATCAGCGACAACG	58	6-FAM	
GBms481	(GT)_13_	F: TGCTGAGCGAGAGAAGCAAT R: CCCAGACTAACACCTCATTTTTATG	58	6-FAM	
MRms245-1 *	(GT)_6_	F: GGTTACTATAAATCTCTGGTGTTACGC R: CCATCTGAGCCACGGTGAAG	58	6-FAM	Present study
MRms245-2 *	(GT)_14_	F: TCGATGCCTACGTGGAGGTC R: ATCATCAGCGTCCGCTCGTA	58	HEX	
MRms245-3 *	(GT)_4_ (GT)_8_	F: AGATGCTCCGACAGATGCGT R: CGCTTCAGAATGAGCCCAGA	58	NED	
MRms637 *	(GT)_21_	F: TGTGAGTTGAGTGCTAACGCTTG R: TCACAAGAGTGAAGGGGTGAATC	58	6-FAM	

* Developed marker. *Ta*: optimal annealing temperature.

**Table 2 genes-14-01611-t002:** Genetic diversity indices estimated for *M. rapidus* from LN, UN, and DC.

Locus		Subpopulation			All
		UN	LN	DC	
PNms172	*n*	48	48	24	120
	*A*	3	3	2	4
	*H* _O_	0.208	0.250	0.083	0.200
	*H* _E_	0.225	0.223	0.082	0.196
	*P* _HWE_	0.017 *	1.000	1.000	0.229
	*F* _IS_	0.077	−0.120	−0.022	−0.022
MKms205	*N*	48	46	24	118
	*A*	7	6	7	8
	*H* _O_	0.813	0.565	0.792	0.712
	*H* _E_	0.702	0.686	0.771	0.708
	*P* _HWE_	0.531	0.172	0.385	0.349
	*F* _IS_	−0.159	0.178	−0.027	−0.003
MKms633	*N*	48	48	24	120
	*A*	21	22	17	25
	*H* _O_	0.896	0.958	0.875	0.917
	*H* _E_	0.938	0.921	0.927	0.932
	*P* _HWE_	0.386	0.417	0.199	0.332
	*F* _IS_	0.045	−0.041	0.058	0.021
MKms747	*N*	48	48	24	120
	*A*	16	15	10	16
	*H* _O_	0.938	0.833	0.792	0.867
	*H* _E_	0.922	0.894	0.812	0.900
	*P* _HWE_	0.099	0.009 **	0.701	0.023 *
	*F* _IS_	−0.017	0.068	0.026	0.026
MRms245-1	*N*	47	46	22	115
	*A*	24	28	14	36
	*H* _O_	0.660	0.674	0.864	0.704
	*H* _E_	0.946	0.945	0.850	0.944
	*P* _HWE_	0.000 ***	0.000 ***	0.389	0.000 ***
	*F* _IS_	0.305	0.289	−0.017	0.192
MRms245-2	*N*	48	47	24	119
	*A*	19	18	13	22
	*H* _O_	0.833	0.894	0.792	0.849
	*H* _E_	0.888	0.907	0.879	0.900
	*P* _HWE_	0.079	0.922	0.138	0.163
	*F* _IS_	0.062	0.015	0.102	0.060
MRms245-3	*n*	48	48	23	119
	*A*	28	28	14	34
	*H* _O_	0.917	0.958	0.957	0.941
	*H* _E_	0.930	0.932	0.895	0.925
	*P* _HWE_	0.518	0.049 *	0.072	0.050
MRms637	*n*	48	46	24	118
	*A*	24	22	12	32
	*H* _O_	0.792	0.761	0.917	0.805
	*H* _E_	0.897	0.846	0.895	0.880
	*P* _HWE_	0.194	0.270	0.174	0.152
	*F* _IS_	0.119	0.101	−0.024	0.065
GBms157	*n*	48	48	24	120
	*A*	36	36	24	44
	*H* _O_	0.896	0.917	0.875	0.900
	*H* _E_	0.966	0.971	0.961	0.967
	*P* _HWE_	0.013 *	0.064	0.024 *	0.001 **
	*F* _IS_	0.074	0.057	0.091	0.074
GBms481	*n*	48	48	24	120
	*A*	5	6	5	6
	*H* _O_	0.667	0.667	0.708	0.675
	*H* _E_	0.678	0.697	0.681	0.685
	*P* _HWE_	0.970	0.243	0.119	0.308
	*F* _IS_	0.017	0.044	−0.041	0.007
Overall	*A*	18.3	18.4	11.8	22.7
	*A*_R_ (*n* = 22)	13.8	13.8	11.5	13.7
	*H* _O_	0.762	0.748	0.766	0.757
	*H* _E_	0.809	0.802	0.775	0.804
	*P* _HWE_	0.000 ***	0.000 ***	0.041 *	0.000 ***
	*F* _IS_	0.056 **	0.053 **	0.000	0.044 **

*A*, number of alleles; DC, Deokcheon River; *F*_IS_, inbreeding coefficient.; *H*_E_, expected heterozygosity; *H*_O_, observed heterozygosity; LN, lower Nam River (Danseong); *n*, number of samples; *P*_HWE_, *p*-value estimated by Fisher’s exact test using the Markov chain method; UN, upper Nam River (Saengcho). * *p* < 0.05, ** *p* < 0.01, *** *p* < 0.001.

**Table 3 genes-14-01611-t003:** Summary Statistics of BOTTLENECK Analysis and Effective Population Size Estimation (*N*_e_) for *M. rapidus* Sampled from UN, LN, and DC.

Subpopulation	BOTTLENECK Tests				*N*_e_ (95% CI)	M-Ratio
	P_IAM_	P_TPM_	P_SMM_	Mode-Shift		
UN	0.009 **	0.903	0.958	L-shaped	175 (93–804)	0.355
LN	0.065	0.998	0.999	L-shaped	157 (79–1077)	0.372
DC	0.016 *	0.935	0.998	L-shaped	61 (29–571)	0.341

CI, confidence interval; DC, Deokcheon River; LN, lower Nam River (Danseong); *N*_e_, estimated effective population size using the LDNe program; P_IAM_, *p*-value in BOTTLENECK test using the infinite allele mutation model; P_SMM_, *p*-value in B_OTTLENECK_ test using the stepwise mutation model; P_TPM_, *p*-value in BOTTLENECK test using the two-phase mutation model (10% variance and 90% proportions of SMM); UN, upper Nam River (Saengcho). *: *p* < 0.05, **: *p* < 0.01.

**Table 4 genes-14-01611-t004:** Pairwise distance and *F*_ST_ calculated among *M*. *rapidus* from UN, LN, and DC using microsatellite genotype analysis.

	UN	LN	DC
UN	-	0.026	0.087
LN	0.001	-	0.082
DC	0.014 ***	0.013 ***	-

Pairwise distance: genetic distance (above), pairwise genetic differentiation: *F*_ST_ (below); DC, Deokcheon River; LN, lower Nam River (Danseong); UN, upper Nam River (Saengcho). *** *p* < 0.001.

**Table 5 genes-14-01611-t005:** Results of Analysis of Molecular Variance (AMOVA) for *M*. *rapidus*.

Source of Variation	d.f.	Sum of Squares	Variance Components	Total Variance (%)	*F*-Statistics
Microsatellite Nam River region vs. Deokcheon River region (UN, LN vs. DC)					
Among groups	1	7.790	0.04772	1.20	*F*_CT_ = 0.012
Among populations within groups	1	4.260	0.00353	0.09	*F*_SC_ = 0.001
Within populations	237	929.396	3.92150	98.71	*F*_ST_ = 0.013 ***
Total	239	941.446	3.97275	100.00	
Nakdong River water system (UN, LN, DC)					
Among groups	2	12.050	0.02739	0.69	*F*_ST_ = 0.007 ***
Within populations	237	929.396	3.92150	99.31	
Total	239	941.446	3.94889	100.00	

***: *p* < 0.001.

## Data Availability

Information of microsatellite markers was published in this study.

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
