# Peer review of "The Impact of Weir Construction in Korea’s Nakdong River on the Population Genetic Variability of the Endangered Fish Species, Rapid Small Gudgeon (Microphysogobio rapidus)"

_genes, 2023, doi:10.3390/genes14081611_

Round 1

Reviewer 1 Report

This paper investigates the effect of weir construction to the population structure of the endangered fish species Microphysogobio rapidus.

According to my opinion and review, I find the work very well done and described and this paper very well written.

Author Response

Thank you for your review.

Reviewer 2 Report

Dear Authors,

The manuscript entitled: “The effects of weir construction in the Nakdong River of Korea on the population genetic viability of the endangered fish, rapid small gudgeon (Microphysogobio rapidus)” represents very interesting and valuable study. The reviewed study aimed to analyze the genetic diversity and structure of the M. rapidus population in the Nam and Deokcheon rivers by employing microsatellite DNA analysis. The obtained genetic data can be utilized to develop an effective conservation strategy for the existing shoal population. However, the overall quality of information presentation is very low, especially description of Material and Methods as well as Results. Language presentation is also very low. The manuscript bears numerous flaws and major improvements are required. All remarks about this have been placed in the attached file.

In conclusion, I do not recommend the manuscript for publication in the present form and major revision should be made by authors. After that the manuscript should be evaluated again. All remarks, questions and fixes were placed in the attached pdf file (yellow highlights contain fixes and sentence suggestions, while red highlights contain comments and questions).

Dear Editors,

Dear Authors,

The manuscript entitled: “The effects of weir construction in the Nakdong River of Korea on the population genetic viability of the endangered fish, rapid small gudgeon (Microphysogobio rapidus)” represents very interesting and valuable study. The reviewed study aimed to analyze the genetic diversity and structure of the M. rapidus population in the Nam and Deokcheon rivers by employing microsatellite DNA analysis. The obtained genetic data can be utilized to develop an effective conservation strategy for the existing shoal population. However, the overall quality of information presentation is very low, especially description of Material and Methods as well as Results. Language presentation is also very low. The manuscript bears numerous flaws and major improvements are required. All remarks about this have been placed in the attached file.

In conclusion, I do not recommend the manuscript for publication in the present form and major revision should be made by authors. After that the manuscript should be evaluated again. All remarks, questions and fixes were placed in the attached pdf file (yellow highlights contain fixes and sentence suggestions, while red highlights contain comments and questions).

Thank you for another interesting manuscript that I could review!

Author Response

Q1.

This table should be replaced by the sampling map. Information about weirs is very unclear here. It would be better to present high quality map with the marks of water flow direction and locations of all the most important weirs. Information about fish number and exact location of theri collection should be also included. I would merge the sampling map with the figure 1 as separate boxes a, b, and c.A1.

A1. Thanks for the review. Edited based on reviewer comments.

Q2.

F prmers were fluorescently labelled? Please more details about this.

A2.

Thanks for the review. Edited based on reviewer comments.

[Table 1]

Q3.

This sentence is very unclear and makes my concern about analysis robustness. What do you mean "if a PCR amplification was not performed" and "or there was difference in the number of individuals"? If PCR conditions for markes amplification are correct the markers should amplify for each sample and every sample should be analyzed by the same marker panel.  Why the number of analysed samples changes?

A3.

Thanks for the review. Edited based on reviewer comments.

In cases where PCR amplification was not performed, small fins of endangered fish were extracted. In this process, it means that the DNA is not amplified due to insufficient amount. Therefore, this case was excluded from the analysis, resulting in a difference in the number of samples analyzed.

[Samples with differences between populations in the number of individuals sampled can be analyzed using FSTAT software ver. 1.2 to estimate the allelic richness in cases where PCR amplification cannot be performed due to insufficient amounts of gDNA from an endangered species [24].]

Q4.

Did not you use Evanno's delta K methods for estimation K?

A4.

Thanks for the review. Estimated using Evanno's delta K. You cannot visualize the result of STRUCTURE without using Evanno's delta K.

Q5.

Please extend this thought. Say more.

A5.

Thanks for the review. Edited based on reviewer comments.

[The effective population size was determined to be 177 (122–296), which is higher than the effective population size of 100 required to maintain the size of the population in the short term but far less than the effective population size of 1000 required over the long term [35]. In conservation genetics, small population size potentially accelerates extinction processes by extinction factors [36]. One important genetic factor that can affect extinction risk at small population sizes is inbreeding depression [35]. Based on this rationale, it can be assumed that the long-term sustainability of M. rapidus is at risk [35, 36].]

Q6.

Unclear, what do you mean?.

A6.

Thanks for the review. Unclear sentences have been deleted.

Q7.

Please improve this sentence. It does not much correspond to the upper one. How low genetic differentiation represents a threat to the survival of the species? How it can lead to loss of genetic diversity?

A7.

Thanks for the review. Edited based on reviewer comments.

[In the case of M. rapidus, there are three sites but only one population. Therefore, although genetic diversity is high, it is likely to become extinct due to habitat destruction or climate change, considering that the population has decreased to the point where it cannot actively respond to environmental changes evolutionarily.]

Q8.

I do not understand this. Provide more detalis.

A8.

Thanks for the review.

[In the case of fish such as Thymallus thymallus, Salvelinus leucomaenis, and Lethenteron sp., it has been reported that manmade structures such as weirs, sluice gates, and dams installed in rivers impede migration [3–5].]

Q9.

This needs to be better explained. Is the Jinyang Lake only barrier separating both rivers?

A9.

Thanks for the review. Edited based on reviewer comments.

[A fishway installed on the right side of the berm cannot be used by the small M. rapidus due to its high slope. Fish migrate frequently during the spawning season, but the spawning season of M. rapidus is in the dry season (4–5 months), such that the low water level will hinder the migration of fish in the Nam River (UN, LN) to those in the Deokcheon River (DC) via Duin weir.]

Q10.

Please explain the observed asymmetrical gene exchange within Nam River and opposite trend between DC and LN also.

A10.

Thanks for the review. Edited based on reviewer comments.

Q11.

Please include/discus in this paragraph the results of gene flow rates analysis.

A11.

Thanks for the review.

[The upstream genetic flow between DC and LN has a flow rate of 29.74%, but upstream flow between LN and DC appears to be almost nonexistent (0.12%). The Duin weir is located between LN and DC. Considering the results of genetic flow analyses, Duin weir (2.0 × 343 × 1.5 m, 1945 year) appears to be blocking the movement of fish, upstream of the shoal. A terraced fishing ground (4.5 × 36 × 1.5 m) was completed on the right side of Duin weir in 2011, but it is estimated that its use is limited. There are many weirs in DC. In particular, the Duin weir is judged to have intensified the blockage of gene flow. To ensure the effective conservation of the species, measures should be taken to facilitate genetic exchange across the study area.]

Round 2

Reviewer 2 Report

I am pleased to inform you that the manuscript entitled "The effects of weir construction in the Nakdong River of Korea on the population genetic viability of the endangered fish, rapid small gudgeon (Microphysogobio rapidus)" has undergone significant improvements. The Authors have diligently addressed all of my remarks and suggestions, resulting in a manuscript of commendable quality. Only minor editorial corrections are now required.

To ensure the highest language presentation standards, I recommend having a native speaker review the text once more. Unfortunately, the current draft is uploaded with tracking changes, making it challenging for me to suggest additional corrections effectively. A final review by a native speaker would undoubtedly enhance the overall clarity and coherence of the manuscript.

Apart from language presentation improvement, I have no further suggestions, and I believe the manuscript is ready for publication after minor corrections.

Best regards,

I am pleased to inform you that the manuscript entitled "The effects of weir construction in the Nakdong River of Korea on the population genetic viability of the endangered fish, rapid small gudgeon (Microphysogobio rapidus)" has undergone significant improvements. The Authors have diligently addressed all of my remarks and suggestions, resulting in a manuscript of commendable quality. Only minor editorial corrections are now required.

To ensure the highest language presentation standards, I recommend having a native speaker review the text once more. Unfortunately, the current draft is uploaded with tracking changes, making it challenging for me to suggest additional corrections effectively. A final review by a native speaker would undoubtedly enhance the overall clarity and coherence of the manuscript.

Apart from language presentation improvement, I have no further suggestions, and I believe the manuscript is ready for publication after minor corrections.

Best regards,

Author Response

Q1. I am pleased to inform you that the manuscript entitled "The effects of weir construction in the Nakdong River of Korea on the population genetic viability of the endangered fish, rapid small gudgeon (Microphysogobio rapidus)" has undergone significant improvements. The Authors have diligently addressed all of my remarks and suggestions, resulting in a manuscript of commendable quality. Only minor editorial corrections are now required.

To ensure the highest language presentation standards, I recommend having a native speaker review the text once more. Unfortunately, the current draft is uploaded with tracking changes, making it challenging for me to suggest additional corrections effectively. A final review by a native speaker would undoubtedly enhance the overall clarity and coherence of the manuscript.

Apart from language presentation improvement, I have no further suggestions, and I believe the manuscript is ready for publication after minor corrections.

Best regards,

Comments on the Quality of English Language

I am pleased to inform you that the manuscript entitled "The effects of weir construction in the Nakdong River of Korea on the population genetic viability of the endangered fish, rapid small gudgeon (Microphysogobio rapidus)" has undergone significant improvements. The Authors have diligently addressed all of my remarks and suggestions, resulting in a manuscript of commendable quality. Only minor editorial corrections are now required.

To ensure the highest language presentation standards, I recommend having a native speaker review the text once more. Unfortunately, the current draft is uploaded with tracking changes, making it challenging for me to suggest additional corrections effectively. A final review by a native speaker would undoubtedly enhance the overall clarity and coherence of the manuscript.

Apart from language presentation improvement, I have no further suggestions, and I believe the manuscript is ready for publication after minor corrections.

A1. Thank you for your review. The manuscript was finally reviewed by native speakers (http://www.textcheck.com/certificate/mkFKhT).